# The Hyperoxic-Hypoxic Paradox

**DOI:** 10.3390/biom10060958

**Published:** 2020-06-25

**Authors:** Amir Hadanny, Shai Efrati

**Affiliations:** 1The Sagol Center for Hyperbaric Medicine and Research, Shamir (Assaf-Harofeh) Medical Center, Zerifin 70300, Israel; efratishai@outlook.com; 2Sackler School of Medicine, Tel-Aviv University, Tel-Aviv 6997801, Israel; 3The Mina and Everard Goodman Faculty of Life Sciences, Bar Ilan University, Ramat-Gan 5290002, Israel; 4The Sagol School of Neuroscience, Tel-Aviv University, Tel-Aviv 6997801, Israel

**Keywords:** hyperoxia, hypoxia, hyperbaric oxygen, biogenesis, hyperoxic-hypoxic paradox, hypoxia-inducible factor (HIF)

## Abstract

Effective metabolism is highly dependent on a narrow therapeutic range of oxygen. Accordingly, low levels of oxygen, or hypoxia, are one of the most powerful inducers of gene expression, metabolic changes, and regenerative processes, including angiogenesis and stimulation of stem cell proliferation, migration, and differentiation. The sensing of decreased oxygen levels (hypoxia) or increased oxygen levels (hyperoxia), occurs through specialized chemoreceptor cells and metabolic changes at the cellular level, which regulate the response. Interestingly, fluctuations in the free oxygen concentration rather than the absolute level of oxygen can be interpreted at the cellular level as a lack of oxygen. Thus, repeated intermittent hyperoxia can induce many of the mediators and cellular mechanisms that are usually induced during hypoxia. This is called the hyperoxic-hypoxic paradox (HHP). This article reviews oxygen physiology, the main cellular processes triggered by hypoxia, and the cascade of events triggered by the HHP.

## 1. Introduction

Oxygen is the third-most abundant element in the universe, after hydrogen and helium, and it is the most dominant effector of most living creatures on earth. About 300 million years ago, during the Carboniferous period, atmospheric oxygen levels reached a maximum of 35%, which may have contributed to the large size of animals and insects at this time [1,2]. Today, oxygen constitutes 20.8% of the earth’s atmosphere, and any slight change in its concertation will have a dramatic impact on all levels of mammalian physiology. The ability to maintain oxygen homeostasis is essential for survival, and all mammalian physiological systems evolved to ensure the optimal level of oxygen supplied to all cells in each organism. This has transpired through the evolution of a complex physiological infrastructure for oxygen delivery (the lungs), oxygen transport carriers (erythrocytes and plasma), oxygen transport pathways (vascular system), and the pump (heart). Both the development and regulation of these systems in organisms provide the basis for oxygen homeostasis.

Effective metabolism is highly dependent on a narrow therapeutic range of oxygen. Accordingly, low levels of oxygen, or hypoxia, are one of the most powerful inducers of gene expression, metabolic changes, and regenerative processes, including angiogenesis and stimulation of stem-cell proliferation, migration, and differentiation. The sensing of decreased oxygen (hypoxia) levels or increased (hyperoxia) oxygen levels occurs through specialized chemoreceptor cells and metabolic changes at the cellular level which regulate the response. Interestingly, in the cellular milieu, fluctuations in free oxygen concentrations rather than the absolute level of oxygen can be interpreted as a lack of oxygen. Thus, intermittent increases in oxygen concentration can induce many of the mediators and cellular mechanisms that are usually induced during hypoxia. This is the so- called hyperoxic-hypoxic paradox (HHP).

In this article, we review oxygen physiology, the main cellular processes triggered by hypoxia, and the cascade of events triggered by the HHP.

## 2. Oxygen Homeostasis

All of our body’s tissues rely on a continuous oxygen supply at a rate that matches the changing metabolic demands. The oxygen delivery chain begins at the lungs, with the destination being the mitochondria (Figure 1). Oxygen is delivered by convection in the airways and blood vessels. It then diffuses across the alveolar-capillary membrane and through the capillary wall to the interstitium and to its destination, the mitochondria. In the blood, oxygen is carried in two forms—a fraction that is bound to hemoglobin and a free fraction dissolved in the plasma. The amount of dissolved oxygen is proportional to the oxygen partial pressure at a specific temperature, according to Henry’s law [3]. At physiologic normoxic conditions, i.e., at normal content of inspired oxygen (20.8%), most (up to 99%) of the oxygen is carried by hemoglobin, and the dissolved quantity of oxygen is small [4]. Therefore, the amount of red blood cells will dominantly affect the total capacity of oxygen delivery. However, at an elevated partial pressure of oxygen (such as breathing pure oxygen and during a hyperbaric exposure), the dissolved amount can become significant. In all cases, the diffusion gradients are oxygen’s driving force from the plasma to the mitochondria. Thus, the free dissolved fraction has a dominant effect on the mitochondria. It is assumed that oxygen freely diffuses across cell membranes. However, recent data indicate that water channels, such as aquaporin-1 (AQP-1), also work as oxygen transporters [5]. 

At the normal steady state, oxygen homeostasis is a dynamically regulated process. The physiological cellular demand for oxygen can vary depending on tissue requirements at a given moment. For example, an exercising muscle has significantly higher oxygen demand than a relaxed muscle. Consequently, the oxygen delivery rate to tissues is continuously adjusted.

Several mechanisms enable the human body to maintain an adequate oxygen supply [6]. The systemic sensing and response to hypoxia include (a) carotid body glomus, a chemosensor in the arterial circulation that can increase the ventilation rate, (b) and the neuroepithelial bodies (NEBs), which are clusters of cells exposed to the airway lumen at branching points, innervated by the Vagus nerve which dilate the pulmonary arteries to optimize the ventilation–perfusion exchange. The adrenergic system is activated during hypoxia, increasing cardiac output and selective blood perfusion to the more crucial organs. At the organ level, arteries dilate in response to hypoxia to increase oxygen delivery. Oxygenation levels are also sensed by the kidneys, which regulate erythropoietin production to adjust the number of red blood cells (RBC) and oxygen-carrying capacity. In addition to the dynamic respiratory and metabolic systems allowing increased oxygen delivery, as outlined above, it is necessary to have regulating mechanisms at the cellular level. These are essential for survival at extreme environmental conditions and pathological/disease states where systemic regulation is insufficient.

At the cellular level, 80% of the available oxygen is used by the mitochondria, while only 20% is used by other organelles. Being the major oxygen consumer and metabolizer, the partial pressure in the mitochondria is very low, only 1–3 mmHg (Figure 1). Mitochondria are, therefore, the key oxygen sensor, as well as important signaling organelles [7]. Most evidence so far points to the following theory: mitochondria signal the onset of hypoxia by generating reactive oxygen species (ROS) signals by the electron transport chain. When ROS are released to the intermembrane space, they interplay with the activation of enzymes, transcription factors, and post-translation responses [8].

In addition, several cells can adapt to reduced oxygen levels by unique mechanisms [6]. For example, the hemoglobin in red blood cells can change its form and affinity under hypoxic conditions [4]. Cytochrome P-450 monooxygenases (CYP) include a vast number of homologous oxygen-sensitive proteins that oxidize a wide range of compounds. Numerous reports have suggested that CYP metabolites contribute to the hypoxia response in the systemic microvasculature and endothelium, and may contribute to hypoxic pulmonary vasoconstriction [6,9,10].

## 3. Hypoxia-Induced Cellular Cascade

As detailed above, effective metabolism is highly dependent on a narrow therapeutic window of oxygen. However, while the arterial oxygen partial pressure in adult mammals is ≈100 mmHg, in the mammalian fetus, it is around 40 mmHg. Thus, hypoxia is a relative term and is most usefully defined as a condition in which failure of either delivery or utilization of oxygen limits normal function.

### 3.1. Hypoxic Inducible Factor

A low level of oxygen is signaled by all living cells, which starts with a class of enzymes called prolyl hydroxylase domain (PHD) proteins [11,12] (Figure 2). Under normal oxygen concentrations, these PHD enzymes include oxygen-sensing hydroxylases which hydroxylate specific proline and asparagine residues on the α-subunit of the transcription factor hypoxia-inducible factor (HIF). Following hydroxylation, the HIF-1α subunit is targeted by the E3 ubiquitin ligase, also known as the von Hippel–Lindau protein (VHLp), which induces HIF-1α ubiquitination and degradation. Factor inhibiting HIF (FIH) is an additional hydroxylase, which hydroxylates an asparagine residue on the α-subunit of HIF, deactivating the HIF transcription factor. HIF is a heterodimer composed of HIF-1α, HIF-2α, or HIF-3α subunits, which dimerize with HIF-1β, HIF-2β, and HIF-3β subunits respectfully forming HIF-1, HIF-2, and HIF-3 factors. HIF-1α is produced in all cell types while the HIF-2α subunit is found in specific cells, including myeloid cells, liver parenchyma, vascular endothelia, type II pneumocytes, and renal interstitium. While HIF-1 and HIF-2 function as lead regulators of the transcriptional response to hypoxia, the HIF-3 function is yet to be known. Although the *HIF1A* gene is constitutively expressed at low levels under normoxic conditions, it is significantly upregulated in response to hypoxia [13].

HIF-1, when stabilized by hypoxic conditions (without hydroxylases repression) (Figure 2), serves as a transcription factor that regulates over 100 genes essential for survival in oxygen-deprived conditions [14]. These include glycolysis enzymes, which allow adenosine triphosphate (ATP) synthesis in an oxygen-independent manner, enzymes decreasing the basal respiratory rate, and upregulating the vascular endothelial growth factor (VEGF) to induce angiogenesis which improves tissue perfusion. Unlike HIF-1, HIF-2 regulates iNOS (inducible NO synthase) and other factors that support increases in tissue oxygenation in adults, such as erythropoietin production [15]. In addition to modulating cellular metabolism needs for survival in a hypoxic environment, HIF-1 can either turn on or off key mammalian regeneration processes [16,17,18]. In controlled mammalian models, it was demonstrated that continuous downregulation of HIF-1 results in a scarring response and loss of tissue [16]. HIF-1 also has an important role in the regeneration and maintenance of essential organ functions that are highly oxygen-dependent such as the brain and the heart. Research conducted on mice demonstrated that increased HIF expression is neuroprotective and enhances regenerative effects that can enhance hippocampal memory and induce better neuroregeneration in post-stroke and spinal cord injuries [17,19,20]. With regard to the heart, increased HIF expression initiates the metabolic processes needed to regenerate the damaged myocardium and improve cardiac functions after various types of injuries [18,21].

### 3.2. Vascular Endothelial Growth Factor (VEGF)

The VEGF family comprises five members: VEGF-A, placenta growth factor (PGF), VEGF-B, VEGF-C, and VEGF-D. VEGF-A is the most known factor out of the VEGF family. All VEGF factors induce different cellular responses by binding to their respective tyrosine kinase receptors located on the cell surface, inducing the receptors’ dimerization, phosphorylation, and activation [22]. VEGF-A production is induced by HIF-1, which in turn activates vascular cells to initiate angiogenesis (the budding of new capillaries from existing vessels) and arteriogenesis (the remodeling of collateral blood vessels to handle the increased flow, bypassing stenotic regions of the original conduit arteries) [11,23]. In addition, the VEGF-A factor induces vasodilatation activity, as well as microvascular permeability increase needed for immediate improvement of tissue ischemia [11]. VEGF-A also stimulates the mobilization of bone marrow-derived angiogenic cells (BMDACs), which in turn migrate to ischemic tissues and participate in angiogenesis and arteriogenesis [11].

### 3.3. Sirtuin

Sirtuins are a family of signaling proteins involved in metabolic regulation. Sirtuins activities include mono-adenosine diphosphate (ADP)-ribosyltransferase or deacylase [24,25]. In mammals, seven sirtuins (SIRT 1–7) have been discovered in different subcellular compartments: SIRT1, SIRT6, and SIRT7 located in the nucleus, SIRT2 in the cytoplasm, and SIRT3, SIRT4, and SIRT5 in the mitochondria [24,25]. SIRT1 is involved in various mechanisms regulating apoptosis, inflammation, and senescence, which are associated with aging-related diseases [14,26,27,28,29]. Reductions in the nuclear energy state and nicotinamide adenine dinucleotide (NAD^+^) levels decrease the SIRT1 activity, which results in diminished pVHL levels and the stabilization of HIF-1α. Conversely, overexpression of SIRT1 promotes mitochondrial biogenesis by deacetylation, resulting in the activation of HIF-1α [30]. The decreased level of SIRT1, in both transcriptional and posttranscriptional stages during aging is considered to be a major metabolic pathway that attenuates mitochondrial biogenesis and causes aging-related diseases [14]. In mice models, overexpression of SIRT1 results in delayed aging phenotypes and lifespan extension, whereas inhibition of SIRT1 annuls the lifespan extension [31]. Overexpression of SIRT1 or SIRT6 is protective in many murine disease models, including cancer, type 2 diabetes, and cardiovascular disease [26,28,32,33,34,35].

The crosstalk between oxygen- and redox-responsive signal transducers occurs through the SIRT1-HIF interaction [36,37]. During hypoxia, the reduced NADH consumption in mitochondria and increased NADH production from glycolysis, causes a reduction of the NAD+/NADH ratio, downregulating SIRT1 transcription. Reduced SIRT1 allows the acetylation of HIF-1α and its activation and provides a positive feedback loop that maintains high levels of HIF-1 activity during hypoxia. Due to the metabolic crisis in hypoxia, HIF-1 blocks mitochondrial energy metabolism by (1) inducing PDK-1, which inhibits the conversion of pyruvate to acetyl-CoA, by (2) inhibiting mitochondrial biogenesis via c-Myc repression, and by (3) reducing mitochondrial transcription factor A (TFAM), required for replication, transcription, and maintenance of mitochondrial biogenesis [38,39].

### 3.4. Mitochondria Biogenesis

Maintaining an adequate mitochondrial population during one’s lifetime is crucial. Mitochondrial quality control occurs via the process of mitophagy, degradation of damaged mitochondria, and generation of newly functioning mitochondria by mitochondrial biogenesis, a process in which new mitochondria are formed by growth and division of preexisting mitochondria. Mitochondrial biogenesis is crucial for preserving most human cell integrity, and in recent years, there is growing evidence about their important role in preserving brain functionality and in neurodegenerative diseases [40].

Mitochondrial biogenesis is a complex process requiring coordinated bi-genomic (cellular and mitochondrial DNAs) regulation to execute several distinct processes, including (1) inner and outer mitochondrial membrane synthesis, (2) synthesis of mitochondrial-encoded proteins, (3) synthesis and import of nuclear-encoded mitochondrial proteins, and 4) replication of mitochondrial DNA (mtDNA). Several cell-signaling pathways tightly regulate mitochondrial biogenesis. The AMP-activated kinase (AMPK)-PGC-1a axis and Sirtuin 1 (SIRT1)-PGC-1a are two major pathways that regulate mitochondrial biogenesis. AMPK can be activated by physiological stimuli such as exercise, starvation, and *transient* hypoxia [41,42,43,44].

There is considerable overlap between the AMPK and HIF signaling pathways, as both are involved in energetic stresses, though the relationship is complex, with both opposing and cooperative outcomes depending on the context [45]. As mentioned, AMPK is a potent stimulator of mitochondrial biogenesis, which in normoxic conditions acts to restore ATP homeostasis. However, as detailed above, once AMPK activation occurs under hypoxic conditions, i.e., energetic stress, additional mitochondrial biogenesis, and oxygen consumption would further decrease oxygen availability and worsen the stress. Therefore, HIF signaling under prolonged hypoxia generally acts to decrease mitochondrial biogenesis and mass [45]. 

### 3.5. Stem Cells

Stem cells (SC) are undifferentiated or partially differentiated cells that can differentiate into various types of cells and divide indefinitely to produce more of the same stem cell. In mammals, the major groups of stem cells include hematopoietic stem cells (HSC), which replenish blood and immune cells, basal cells, which maintain the skin epithelium, and mesenchymal stem cells (MSC), which maintain bone, cartilage, muscle and fat cells. There are also more targeted subgroups of SCs that include neuronal stem cells (NSC) that persist in restricted regions in the adult brain and continue to produce neurons throughout the person’s life. NSCs can generate nerve cells and their supporting cells, oligodendrocytes, and astrocytes [46]. In addition to their differentiation capacity, SCs have multipotent properties that include anti-inflammatory effects that make them potential therapeutic candidates for a variety of disorders [47]. 

During normal conditions, SCs are at a reversible state of quiescence, i.e., at a temporary cell cycle arrest [48]. While being quiescent, they exhibit improved stress resistance and enhanced survival ability. Oxygen has an important role in the regulation of stem cell proliferation and differentiation [48,49,50]. Short term hypoxia can induce SC proliferation, migration, and differentiation capacity [48,50]. In addition, hypoxia also modulates the paracrine activity of MSCs, causing upregulation of various secreted factors, such as VEGF and exosomes, that also have important angiogenesis and anti-inflammatory effects [50,51,52]. The mechanisms by which hypoxia exerts its effect on cells is mainly regulated by HIF-1 and related down-cascade protein expression [48,50]. 

## 4. The Hyperoxic Hypoxic Paradox

As detailed above, hypoxia is the natural trigger for mitochondria metabolic changes via elevated levels of HIF, VEGF, Sirtuin, mitochondria metabolic changes, and SC proliferation, and migration. However, the understanding that at the cellular level, oxygen level fluctuations can trigger a cellular cascade that is usually triggered by hypoxia, allows the use of intermittent hyperoxia to stimulate tissue regeneration without the hazardous effects of hypoxia. This is termed the “hyperoxic-hypoxic paradox” (Figure 3).

In clinical practice, intermittent hyperoxia can be generated using hyperbaric oxygen therapy (HBOT). HBOT includes inhaling 100% oxygen at pressures exceeding 1 atmosphere absolute (ATA) to enhance the amount of oxygen dissolved in the body tissues. During HBOT, the arterial O_2_ tension typically exceeds 1500 mmHg, and levels of 200–400 mmHg occur in tissues. In a normal individual, at a normal environment (20.8% oxygen at 1ATA), the hemoglobin is almost entirely saturated (94–99%). Accordingly, when in a hyperbaric environment, the effect is only in the dissolved oxygen. As mentioned above, the dissolved oxygen is the fraction responsible for the diffusion gradient from the capillaries to the mitochondria. In the following part of the article, we will review the cellular cascade induced following repeated transient hyperoxia. 

### 4.1. Hypoxia-Inducible Factor

It has been proposed that the relative changes in oxygen availability rather than constant hypoxia or hyperoxia have a more dominant effect on HIF expression [53,54,55]. According to this hypothesis, the cells interpret the change from normoxia to hypoxia or the change back to normoxia following a hyperoxic exposure as an oxygen shortage and induce HIF-1-regulated gene synthesis [53,54,55]. Even though there is a growing understanding of the cellular cascade responsible for HIF expression (as detailed below), the precise level of inspired oxygen and the exact timeframe for its iterative administrations are not fully known yet.

The proposed mechanisms for increased HIF availability at normoxia after hyperoxic exposure relates to ROS availability and scavengers, such as the glutathione synthase and superoxide dismutase (SOD) enzymes [53,54,55]. As detailed above (Figure 2), at normoxic conditions when oxygen and its ROS derivates are available at higher ratios compared to their scavengers, most of the HIF-1α subunits are hydroxylated by PHD. This results in their ubiquitination by VHLp, and consequently, degradation of HIF-1α in the proteasome. In hypoxia, oxygen and its ROS derivates are less available, the HIF-1α is not hydroxylated and ubiquitinated by VHLp, and more HIF-1α is available to enter the nucleus and is dimerized with HIF-1β to form the active HIF promotor. During hyperoxia, increased levels of oxygen availability will enhance the production ROS and the production of ROS scavengers, including glutathione peroxidase, SOD, as well as other ROS scavengers [56,57,58,59,60]. Based on the above cumulative data, our perspective is that this scavenger level increase is limited and gradual following a single hyperoxic exposure, and repeated exposures are required in order to obtain significant scavenging activity (Figure 2). After returning to normoxia from a single hyperoxic exposure, the ROS/scavenging capacity ratio will be high, leading to ubiquitination of all HIF molecules. Upon return to normoxia following repeated hyperoxic exposures, the level of scavengers is increased, along with their inherent elimination half-life (T_1/2_), which is significantly longer than ROS T_1/2_. Thus, the ratio of ROS/(scavenging capacity) will be low, similar to the hypoxic state. With enhanced scavenging capacity, less ROS is available to bind PHD, less degradation of HIF-α subunits, which, in turn, will enter the nucleus to bind to HIF-1β and generate the active HIF gene promotor. Meaning, intermittent hyperoxia generates a hypoxia-mimicking state without hypoxia by decreasing the ratio of ROS/scavenging capacity.

The effect of repeated intermittent hyperoxia by HBOT on HIF expression was demonstrated in different types of organs and cells [57,58,61,62]. For example, HBOT can induce HIF expression and its related regenerative cascade in the injured brain [58,63], gastrointestinal tract [57], and liver [61,62]. In addition, HBOT induces the expression of different types of HIF and in different types of stem cells [59,64]. The dose-response curve related to the applied pressure, time, and number of HBOT exposures and its relation to HIF expression is still not fully understood, and further studies are needed to find the optimal HBOT protocols. It should be noted that when HBOT is administrated to an ischemic tissue, which overexpresses HIF, tissue hypoxia is being reversed and corrected, accordingly the overexpressed HIF is reduced towards the normalized baseline [65,66,67,68,69,70]. Moreover, when HBOT is being used as preconditioning therapy for hypoxia (by HIF induction), the tissue will tolerate the ischemic insult better, less ischemic injury induced, and the overall post insult HIF expression will be lower than expected for the same insult [71,72].

### 4.2. VEGF and Angiogenesis

VEGF production is induced by HIF-1 and then goes on to stimulate the cellular processes needed for both angiogenesis and arteriogenesis (as detailed above). VEGF is significantly increased following intermittent hyperoxic exposures, utilizing the HHP. There is growing evidence from preclinical as well as from clinical studies demonstrating that repeated HBOT sessions induce the crucial elements for angiogenesis, VEGF expression, and endothelial progenitor cells (EPCs) [61,73,74,75]. Unlike VEGF induced under ischemic conditions, VEGF induction under hyperoxic stimuli can facilitate angiogenesis in tissues that are hypoxic/ischemic while breathing normal air. Clinical studies have confirmed that repeated daily HBOT sessions augment the circulating levels of VEGF, EPCs, and improve the blood flow in ischemic areas of patients with chronic peripheral arterial occlusive diseases, with or without non-healing wounds [73,74,75,76]. 

The use of hyperoxic stimuli to induce angiogenesis is crucial for organs that are oxygen- deprived, even at normal conditions such as the human brain [77,78]. Therefore, to enable brain angiogenesis and regenerative processes, it is essential to increase oxygen delivery in addition to VEGF induction, which is achieved by cyclic hyperoxic exposure. HBOT can initiate the cellular and vascular repair mechanisms to induce brain angiogenesis and improve cerebral blood flow in damaged brain regions [79,80,81].

### 4.3. Sirtuin

SIRT1 acts as a metabolic sensor by its ability to deacetylate the mitochondrial biogenesis factor PGC-1α. During hyperoxia, the NAD+/NADH ratio is increased through the Krebs cycle and mitochondrial metabolism, triggering SIRT1 [82,83]. SIRT1 is known to deacetylate FOXO3a, which induces antioxidant responses via modulating SOD2 and CAT. FOXO3a has further been shown to regulate mitochondrial gene expression, resulting in modulated ROS levels [82]. SIRT1 regulates the acetylation of PGC-1α, a master regulator of mitochondrial biogenesis [83]. SIRT1, along with other sirtulins appears to be an integral part of an important cellular defense mechanism against oxidative stress and ROS formation.

Intermittent hyperoxic exposures, can produce ROS molecules that increase the activity of SIRT-1 inside cells through the activation of mitogen-activated protein kinase (MAPK) [84]. The effect of intermittent hyperoxic exposure was evaluated in several animal models. Yan et al. evaluated the effect of HBOT on SIRT1 in a model of focal cerebral ischemia induced by middle cerebral artery occlusion and also on primary cultured cortical neurons subjected to oxygen-glucose deprivation injury [84,85]. Their findings indicate that HBOT’s neuroprotective effect is mediated by increased SIRT1 mRNA and protein expression. HBOT’s neuroprotection was attenuated by a SIRT1 inhibitor and also in SIRT1 knockdown rats [84]. In another study done in hyperglycemic rats after middle cerebral artery occlusion, HBOT induced activation of ATP/NAD and SIRT1 and SIRT2 pathways resulting in attenuation of hemorrhagic transformation, brain infarction, as well as neurological function improvement [86]. Similar results, indicating that the neuroprotective effects of HBOT are mediated by SIRT1 were demonstrated in middle-aged mice with postoperative cognitive dysfunction [87].

### 4.4. Mitochondria

Any change in the free dissolved oxygen generates a diffusion gradient that directly affects the oxygen delivered and sensed by the mitochondria through the production of ROS signaling molecules. Therefore, intermittent increases in dissolved oxygen generated by HBOT can be compared to “intense interval training” with an expected cumulative effect along with repeated exposures. In a well-designed mice model study, it was demonstrated that adding intermittent hyperbaric exposure to exercise training further improves endurance performance by facilitating oxidative and glycolytic capacities and by increasing the expression of proteins involved in mitochondrial biogenesis in striated muscles [88]. In humans, combining HBOT to an exercise training regimen induces better cardiorespiratory fitness compared to exercise training alone [89]. 

Regarding the brain, ROS are considered to have significant metabolic effects, and the main source of ROS in the brain is the mitochondria [90]. Normally 1–2% of the oxygen consumed by the mitochondria is converted to O_2_^−1^ and related ROS metabolites such as hydrogen peroxide (H_2_O_2_) and peroxynitrite (ONOO-) [91]. O_2_^−1^ and related ROS formations increase in proportion to the partial pressure of oxygen, and in high pressure multiplied by the exposure duration. Exposure may induce high neurological activity presented in the form of epileptic seizures [90]. In a study done by Gutsaeva et al., it was demonstrated that pre-convulsive dosages of HBOT induce significant mitochondrial biogenesis in the hippocampus [91]. The cellular response in the hippocampus was mediated by increased *NRF-2* gene expression, *TFAM* gene expression, and mitochondrial gene transcription [91].

In recent years, there is growing evidence about the possibility and the importance of mitochondrial transfer between astrocytes and neurons for proper maintenance of neuronal function and as cell-cell signaling [92,93]. Neurons can release and transfer damaged mitochondria to astrocytes for disposal and recycling [92], and astrocytes can release functional mitochondria that enter into neurons [93]. In a study done by Borlongan and Lippert, it was demonstrated that HBOT could facilitate the transfer of resilient mitochondria from astrocytes to neuronal cells that are more susceptible to inflammation [94]. The mitochondrial transfer from astrocytes to neurons makes the neurons more resilient to inflammatory insults. These findings suggest a new mitochondrial mechanism of neuroglial crosstalk that may contribute to endogenous neuro-protective and neuro-recovery mechanisms induced by HBOT.

### 4.5. Stem Cells

Both hypoxia and intermittent hyperoxia increase HIF and its downstream gene expression, including stem cell factors (SCFs) [95]. Growing data from preclinical and clinical studies demonstrate the cumulative effect of repeated intermittent hyperoxia by HBOT on proliferation and mobilization of stem cells [59,96,97,98,99,100,101,102,103,104,105,106,107,108,109,110,111,112]. Clinical studies on patients suffering from diabetic wounds and post-traumatic brain injury have demonstrated that repeated HBOT sessions increase circulating (mobilization) stem cells in correlation with the clinical improvements [96,97,98].

With regard to subtypes of stem cells, it was demonstrated that HBOT promotes neuronal stem cell proliferation [99,100,101,102,103,104,105], stimulates vasculogenic stem cell growth and differentiation [59,106], stimulates colonic stem cells and induces mucosal healing [107], improves the osteogenic properties of mesenchymal stem cells [108,109], and increases the myoblast growth rate and enhances muscle regeneration [110,111,112].

The main advantage of stimulating stem cells by intermittent hyperoxia instead of hypoxia, is that stem cell proliferation and differentiation, similar to any other regenerative process, is energy-dependent and cannot be accomplished in a hypoxic environment. The fact that oxygen is crucial for stem cells related to regenerative effects has been validated in different studies that compared the results of stem cell injections along with or without HBOT. The potential added value of using HBOT in addition to stem cell injections was seen in a variety of tissues, including brain [113,114], spinal cord and peripheral nerves [115,116], myocardium [117,118], and diabetic wounds [119]. In all those studies, adding HBOT to stem cell injections had synergistic beneficial effects.

### 4.6. Oxygen Toxicity

Although oxygen therapy is considered to be safe, like other active ingredients or drugs, at high dosage, it can be harmful and result in oxygen toxicity. Prolonged exposure to high oxygen pressure with a prolonged imbalance between ROS to scavengers can lead to membrane lipid peroxidation and enzyme inhibition and modulations, most commonly seen in the central nervous system (CNS), that lead to alterations in neuronal metabolism and its related electrical activity [120]. As was first suggested in 1878, breathing hyperbaric oxygen can culminate in grand mal seizures [121]. Another organ that is relatively sensitive to oxygen toxicity is the lung. Pulmonary oxygen toxicity can be manifested by chest tightness, cough, and a reversible decline of pulmonary function [122].

Both CNS and pulmonary toxicity depend upon the partial pressure of oxygen and the duration of exposure [123]. Accordingly, the new HBOT protocol used today includes repeated daily sessions limited to 60–90 min with oxygen partial pressure not exceeding 2.4 ATA, as well as air brakes every 20–30 min. Using those new protocols, HBOT is considered to be safe, while both pulmonary and oxygen toxicity are very rare [124,125,126]. In a recent analysis of 62,614 hyperbaric sessions, the overall incidence of seizures during hyperbaric sessions was 0.011% (1:8, 945 sessions) [124,125]. In addition, in patients without chronic lung diseases, the currently used HBOT protocols do not cause any pulmonary toxicity or changes in pulmonary functions following 60 repeated exposures [126].

## 5. Summary

As in Albert Einstein’s “theory of relativity” that explains the basic physical aspects of our cosmos, relatively can also be found in the micro-cosmos, i.e., the microenvironmental interpretation at the cellular level. As summarized in Figure 3, most of the cellular cascades initiated by hypoxia can be induced by intermittent hyperoxia, the so-called “hyperoxic-hypoxic paradox”. HIF, VEGF, SIRT, mitochondrial biogenesis, and stem cell proliferation and migration could all be induced by “biological fooling” the cells with certain protocols of repeated intermittent hyperoxia. Even though the exact dose response-curve has yet to be discovered in clinical practice, certain HBOT protocols have already demonstrated induction of damaged tissue regeneration.

## Figures and Tables

**Figure 1 biomolecules-10-00958-f001:**
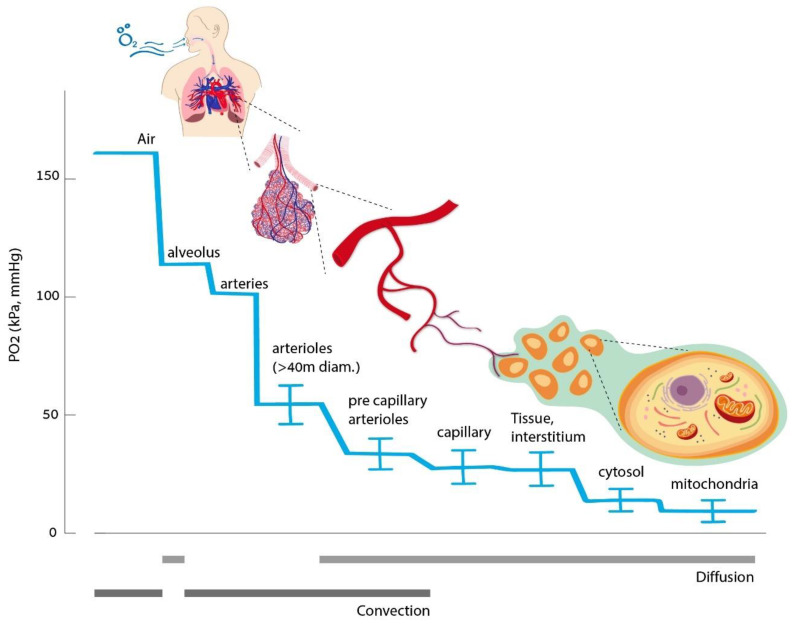
Oxygen delivery chain.

**Figure 2 biomolecules-10-00958-f002:**
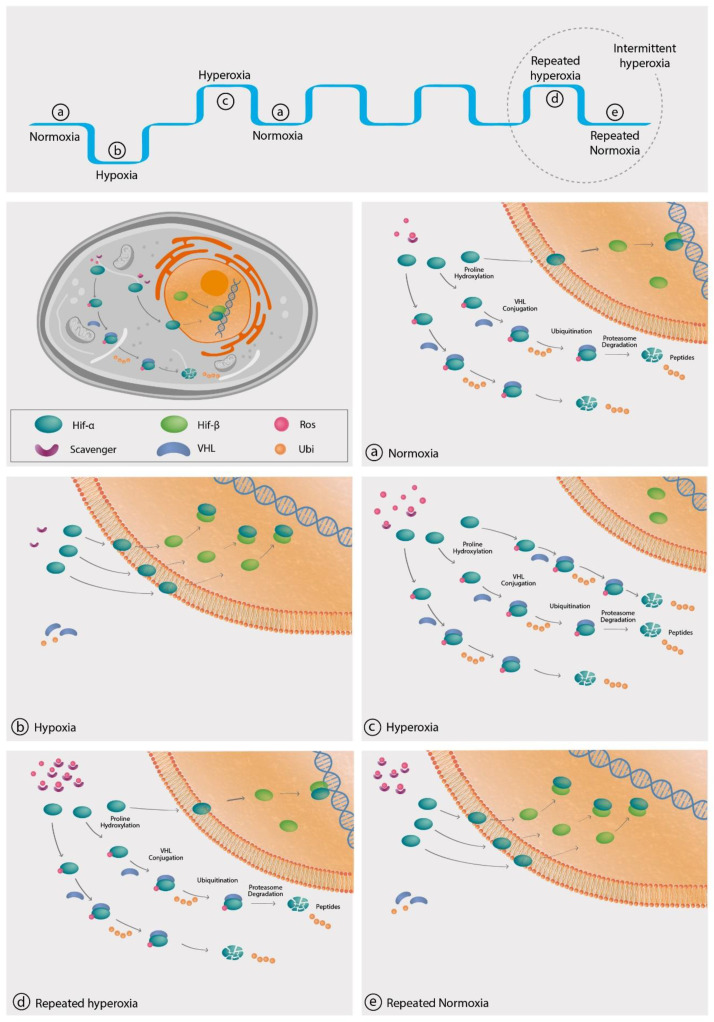
The intracellular cascade of HIF-1 alpha. Legend: HIF-1 is a heterodimer composed of cytoplasmatic HIF-1α and the nuclear HIF-1β subunits. (**a**) Under normal oxygen environments, the ratio of ROS/scavenger is high and the free ROS molecules initiate HIF-1α hydroxylation, HIF-1α subunits become a target for VHLp (von Hippel–Lindau protein) protein which facilitates HIF-1α subunits ubiquitination and degradation. (**b**) Under hypoxic conditions, less oxygen and ROS molecules are available, HIF-1α subunits are not hydrolyzed, and more HIF-1α subunits penetrate the nucleus to conjugate with HIF-1β subunits and generate the active HIF transcription factor. (**c**) At the hyperoxic environment, more ROS and oxygen are available; thus more HIF-1α subunits are hydrolyzed and degraded. (**d**) The adaptive response to repeated hyperoxia includes increases in the production of scavengers that adjust to the increased ROS generation. Thus, the ROS/scavenger ratio gradually becomes similar to the ratio under normal oxygen environment prior to initiating repeated hyperoxic exposures. (**e**) Upon return to normoxia, following repeated hyperoxic exposures, the ratio of ROS/scavenger is low due to the fact scavengers elimination half-life (T_1/2_) is significantly longer than the T_1/2_ of ROS. Accordingly, less HIF-1α subunits are hydroxylated, and more of them penetrate the nucleus, conjugate with HIF-1β to generate the active HIF, similar to the hypoxic state.

**Figure 3 biomolecules-10-00958-f003:**
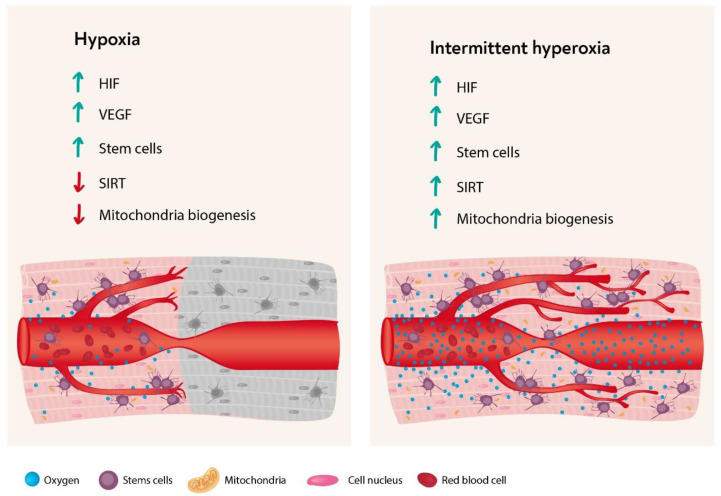
The major cellular response cascade initiated by hypoxia and by intermittent hyperoxia. Legends: HIF: Hypoxic induce factor; VEGF: Vascular endothelial growth factor; SIRT: Sirtuin.

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
