# Peer review of "The Hyperoxic-Hypoxic Paradox"

_biomolecules, 2020, doi:10.3390/biom10060958_

Round 1

Reviewer 1 Report

This article reviews oxygen physiology and the main cellular processes triggered by hypoxia and hyperoxic-hypoxic paradox (HHP), suggesting the neuroprotective effect of hyberbaric oxygen therapy (HBOT), and its potential therapeutic use in other diseases. 

Questions: 

  1. Though intermittent hyperoxia generates a hypoxia-like state without hypoxia, such as increased accumulation of HIF, upregulation of VEGF, the vascular responses to hypoxia and hyperoxia can be different, such as changes in cerebral blood flow (CBF). This should be addressed. 
  2. The disadvantages of HBOT need to be discussed too. The oxygen toxicity during the hyperoxic episode can be harmful. The ROS/scavenging capacity may depend on the duration and the magnitude of hyperoxia. How to design the interval of hyperoxia/normoxia and choose the optimal dose of hyperoxia is  important to maintain cellular oxygen homeostasis. 
  3. If the HBOT is used as a systemic treatment, what is the effect on other organ systems? 

Author Response

Biomolecules

Reply to comments Manuscript   biomolecules-832237

Dear Editor,

Attached, please find a revised version of our manuscript entitled: " The Hyperoxic-Hypoxic Paradox.  

We appreciate your and the reviewer’s efforts in handling this manuscript. Below you will find our reply to the reviewers’ comments one by one.

We hope that you will find the revised version suitable for publication.

Sincerely,

Amir Hadanny, MD

Shai Efrati, MD

Sagol Center for Hyperbaric Medicine and Research,

Shamir Medical Center, 70300 Zerifin, Israel.

Sackler School of Medicine and Sagol School of Neuroscience,

Tel-Aviv University, Tel-Aviv, Israel

Reviewer 1 Comments:

Comment #1: "Though intermittent hyperoxia generates a hypoxia-like state without hypoxia, such as increased accumulation of HIF, upregulation of VEGF, the vascular responses to hypoxia and hyperoxia can be different, such as changes in cerebral blood flow (CBF). This should be addressed."

Reply:  

Indeed, the brain tissue relies on an adequate supply of blood to match oxygen and glucose delivery with the local metabolic demands of active neurons. The cerebral vasculature responds any fluctuation in arterial oxygen pressure by  either vasoconstriction or vasodilatation. For example:

  • Acute exposure to intermittent hypoxia causes vasodilatation and an immediate increase in cerebral blood flow (CBF) which is curvilinear with respect to arterial PO2 and linear with respect to hemoglobin saturation. The exact opposite occurs during intermittent hyperoxia.
  • The acute increase in cerebral blood flow in response to hypoxia can be defined by both the magnitude of change as well as the rate at which this change occurs (i.e. time-course). The CBF response to an immediate hypoxia occurs with an on-transient time constant of ~80 seconds. A difference on transients is observed during hyperoxia. The on-transient response for an immediate isocapnic hyperoxia occurs with a time constant of ~30 seconds. Together, these data suggest that the cerebral vasculature is more responsive to oxygen increase than decreases , regardless of the initial oxygen level.
  • The mechanisms contributing to the dynamic pattern of cerebrovascular acclimatization to prolonged hypoxia include changes in cerebro-spinal fluid (CSF) pH, angiogenesis, and increased hematocrit. Over the first few days to weeks of altitude exposure it is hypothesized that a reset of the relationship between carbon dioxide pressure (PaCO2) and pH occurs in part due to a decrease in CSF fluid bicarbonate (HCO3)-. This reset is thought to result in an increase in CSF pH and a reduced cerebral blood flow. With extended stays at altitude (weeks-to-years) there is an increase in hematocrit, while rodent data also demonstrate an increase and brain capillary density. Both of these adaptations can be expected to improve both convective oxygen delivery and cerebral blood volume.
  • Chronic intermittent periods of hypoxia, as experienced with sleep breathing disorders, are associated with negative effects on cerebrovascular function. Although basal CBF appears unaffected, the cerebrovascular response to changes in blood gases in fact impaired in obstructive sleep apnea (OSA) patients, as well as in normal subjects exposed to prolonged periods of intermittent hypoxia. The brain’s ability to buffer oscillations in blood pressure and maintain constant perfusion (autoregulation) is also impaired in OSA patients and may increase the risk of ischemia during drops in blood pressure. Although the specific mechanism contributing to decreased cerebral function with exposure to chronic intermittent hypoxia remains unknown, there is good evidence suggesting augmented sympathetic vasomotor activity, decreased endothelial function, and increased arterial stiffness are contributing factors.

As detailed above the vasodilatation and vasoconstriction effect of changes in arterial PO2 are significant and important. However, the exact CBF complex responses are behind the scope of this article which focuses on the cascade that ocurrs “universally”, throughout the different organs, in response to hypoxia and hyperoxia (. We agree that the brain CBF response to changes in oxygenation is very interesting and a dedicated article focusing on this topic can be considered for the journal.    

References:

“Influence of Hypoxia on Cerebral Blood Flow Regulation in Humans.” Steinback CD, Poulin MJ. Adv Exp Med Biol. 2016;903:131-44. doi: 10.1007/978-1-4899-7678-9_9.

“Human brain blood flow and metabolism during isocapnic hyperoxia: the role of reactive oxygen species.” Mattos JD, Campos MO, Rocha MP, Mansur DE, Rocha HNM, Garcia VP, Batista G, Alvares TS, Oliveira GV, Souza MV, Videira RLR, Rocha NG, Secher NH, Nóbrega ACL, Fernandes IA. J Physiol. 2019 Feb;597(3):741-755.

“Effect of Hyperoxia on Cerebral Blood Flow Velocity and Regional Oxygen Saturation in Patients Operated on for Severe Traumatic Brain Injury-The Influence of Cerebral Blood Flow iutoregulation.” Sahoo S, Sheshadri V, Sriganesh K, Madhsudana Reddy KR, Radhakrishnan M, Umamaheswara Rao GS. World Neurosurg. 2017 Feb;98:211-216.

Comment #2: "The disadvantages of HBOT need to be discussed too. The oxygen toxicity during the hyperoxic episode can be harmful. The ROS/scavenging capacity may depend on the duration and the magnitude of hyperoxia. How to design the interval of hyperoxia/normoxia and choose the optimal dose of hyperoxia is  important to maintain cellular oxygen homeostasis. 

Reply: we have added a section discussing oxygen toxicity.

Comment #3: "If the HBOT is used as a systemic treatment, what is the effect on other organ systems? "

Reply: The aim of the article was to review the cellular cascade induced by the so called “Hyperoxic- Hypoxic Paradox”. As detailed in the manuscript the induction of HIF, VEGF, stem cells proliferation is relevant to most essential organs in the human body where a chronic  injury/wound occurs.

Additional articles in the upcoming special journal will focus on specific organs and conditions.  

Reviewer 2 Report

Well organized. Figures are clear

Author Response

None

Reviewer 3 Report

This review on the subject ”the Hyperoxic-Hypoxic Paradox” is important as it joints together an increasingly large body of scientific evidence on the importance of oxygen as a key regulatory mechanism in many cellular process. Oxygen sensing is important for many biochemical processes and regulatory mechanisms with therapeutic potential. Overall, I find the manuscript well-written and concise. I have some minor and one major comments to the current form of the manuscript;

Figure 2: I would ease the reading of the figure if the color difference between HIF-a and HIF-b where more clear, i.e. with more contrast (they look very similar so one has to check carefully on the computer screen).

For figures C, D and E I suggest you for example put in more ROS figures in order to graphically demonstrate your point for each different step.

Line 262: separate the words “exposure as”

Line 286-292:

276 – are there any scienfitic references that may confirm the statement “and requires repeated exposures in order to obtain a significant scavenging activity “ ? or do you simply suggest this is how it works ? state so.

One can quickly identify from 10-15 scientific reports describing how intermittent hyperbaric hyperoxia can induce HIF-1a upregulation and subsequent target gene expression. Similarly, there are also easily identified from 10-15 reports describing how hyperbaric hyperoxia can induce HIF-1a downregulation. May I suggest the authors pay more attention to the circumstances of when and under which conditions hyperbaric hyperoxia will induce either the one (upregulation) or the other (downregulation). These findings do not necessarily need to be contradictory. It is likely more a matter of the sequence of the events that lead to one or the other but it would be of importance to the reader to get a more firm grasp on that part of the literature to increase understanding of the dynamics in oxygen sensing when induced by intermittent, hyperbaric hyperoxia.

HIF-downregulation:

JAP 101:853-865, 2006

Pancreas 38:515-522, 2009

Int J Mol Med 39:1299-1306, 2017

J If Investigative Dermatology 128:2102-2112, 2008

UHM, 44(4):321-333, 2017

Stroke 39:1000-1006, 2008

Pediatric Diabetes:11:471-478, 2010

J of Zhejiang Univ-Science B

Stroke:40:3139-3142, sept. 2009

J Cerebral Blood Flow and Metabol:25;554-571, 2005

Cancer Medicine:5(11):3147-3155, 2016

FASAB J; 32:4470-4481, 2018.

Chin Med J:126(20):3897-3903, 2013

J Orthop Res:31:1839-1844, 2013

Medicine 97:29(e11529), 2018

Brain Research:1153:204-213, 2007.

Oman Med Journal;35:1:e90, 2005

Exp Neurology;191:198-210, 2005

HIF-Upregulation:

Wound Repair Regen:19(2):149-161, 2011

Int J Neuroscience, 125:8:625-634

UHM;46 (1):35-44, 2019

Brain Research;1212;71-78, 2008 ¤) unethical study -

J Appl Physiol:104:1185-1191, 2008

Stroke;45:1807-1814, 2014

Neurotox Res;15:367-380, 2009

Wound rep Reg;23:98-103, 2015

J of the Neurological Sciences, 342;114-123, 2014

Academic emergency medicine 2006; 13:707–714

J Appl Physiol 106: 711–728, 2009

Author Response

Biomolecules

Reply to comments Manuscript biomolecules-832237

Dear Editor,

Attached, please find a revised version of our manuscript entitled: " The Hyperoxic-Hypoxic Paradox.

We appreciate your and the reviewer’s efforts in handling this manuscript. Below you will find our reply to the reviewers’ comments one by one.

We hope that you will find the revised version suitable for publication.

Sincerely,

Amir Hadanny, MD

Shai Efrati, MD

Sagol Center for Hyperbaric Medicine and Research,

Shamir Medical Center, 70300 Zerifin, Israel.

Sackler School of Medicine and Sagol School of Neuroscience,

Tel-Aviv University, Tel-Aviv, Israel

Reviewer 3 comments:

This review on the subject ”the Hyperoxic-Hypoxic Paradox” is important as it joints together an increasingly large body of scientific evidence on the importance of oxygen as a key regulatory mechanism in many cellular process. Oxygen sensing is important for many biochemical processes and regulatory mechanisms with therapeutic potential. Overall, I find the manuscript well-written and concise. I have some minor and one major comments to the current form of the manuscript;

Comment #1: “Figure 2: I would ease the reading of the figure if the color difference between HIF-a and HIF-b where more clear, i.e. with more contrast (they look very similar so one has to check carefully on the computer screen).”

Reply: Done

Comment #2: “For figures C, D and E I suggest you for example put in more ROS figures in order to graphically demonstrate your point for each different step.”

Reply: Done

Comment #2: Line 262: separate the words “exposure as”

Reply: Done

Comment #1: "276 – are there any scienfitic references that may confirm the statement “and requires repeated exposures in order to obtain a significant scavenging activity “ ? or do you simply suggest this is how it works ? state so."

Reply: It is our understanding and perspective based on the cumulative data detailed in the manuscript. A statement was added to the manuscript. These are scientific references that have shown the significant effects of repeated exposures:

The Effects of Hyperbaric Oxygen at Different Pressures on Oxidative Stress and Antioxidant Status in Rats

Mn-SOD and Bcl-2 Expression After Repeated Hyperbaric Oxygenation

https://pubmed.ncbi.nlm.nih.gov/21391785/

Comment #2: “One can quickly identify from 10-15 scientific reports describing how intermittent hyperbaric hyperoxia can induce HIF-1a upregulation and subsequent target gene expression. Similarly, there are also easily identified from 10-15 reports describing how hyperbaric hyperoxia can induce HIF-1a downregulation. May I suggest the authors pay more attention to the circumstances of when and under which conditions hyperbaric hyperoxia will induce either the one (upregulation) or the other (downregulation). These findings do not necessarily need to be contradictory. It is likely more a matter of the sequence of the events that lead to one or the other but it would be of importance to the reader to get a more firm grasp on that part of the literature to increase understanding of the dynamics in oxygen sensing when induced by intermittent, hyperbaric hyperoxia". Reply: Thanks for the references. The discussion and references on HIF-1a were further extended. It was noted, with perspective related to the relevant references, that when HBOT is administrated to an already ischemic tissue, which overexpressed HIF, when tissue hypoxia is being reversed and corrected,the over expressed HIF is reduced towards the normalized baseline. Moreover, when HBOT is being used as preconditioning therapy for hypoxia (by HIF induction) the tissue tolerates the ischemic insult better, less ischemic damage, and overall post insult HIF expression will be lower than expected for the same insult.